# REVISITING PROMPT-BASED METHODS IN CLASS IN-CREMENTAL LEARNING

## ABSTRACT

In recent years, prompt-based methods have emerged as a promising direction for continual learning, demonstrating impressive performance across various benchmarks. These methods create learnable prompts to infer task identity, then select and integrate specific prompts into the pretrained model to generate instructed features for prediction. In this paper, we first analyze the working patterns of such method across different distribution scenarios through extensive empirical analysis. Our analysis exposes the limitations of existing methods: first, two-stage inference can make mistakes even when the first stage has already provided reliable predictions; second, enforcing identical architectures for both stages hampers performance gains. To address these issues, we incorporated a self-supervised learning objective to learn discriminative features, thereby boosting the plasticity of the model. During inference, we implemented a simple yet effective threshold filtering strategy to selectively pass data to the second stage. This approach prevents errors in the second stage when the first stage has already made reliable predictions, while also conserving computational resources. Ultimately, we explore utilizing self-supervised pretrained models as a unified task identity provider. Comparing to state-of-the-art methods, our method achieves comparable results under in-distribution scenarios and demonstrates substantial gains under out-of-distribution scenarios (e.g., up to 6.34% and 5.15% improvements on Split Aircrafts and Split Cars-196, respectively).

## 1 INTRODUCTION

Continual learning (CL) aims to equip models with the ability to constantly learn new knowledge without forgetting previously acquired knowledge, with the main challenge being how to mitigate catastrophic forgetting McCloskey & Cohen (1989); Goodfellow et al. (2013) occurs in deep neural networks. Catastrophic forgetting refers to a phenomenon in which introducing new information causes the model to forget old knowledge, thereby drastically reducing its performance on previous learned tasks. Early studies Zenke et al. (2017); Aljundi et al. (2018); Chaudhry et al. (2018) generally start with neural networks that are randomly initialized, focusing primarily on class incremental learning (CIL) scenario Van de Ven & Tolias (2019); De Lange et al. (2021); Masana et al. (2022); Wang et al. (2024b), where each incremental stage involves non-overlapping categories. With the emergence of large-scale pretrained models Dosovitskiy (2020); Kolesnikov et al. (2020); Yalniz et al. (2019); Caron et al. (2021); Oquab et al. (2023), extensive efforts have started to apply them to CL. Among these, a series of works Wang et al. (2022b;a); Smith et al. (2023); Chen et al. (2023); Wang et al. (2024a) are based on prompt learning, which learns and retrieves task-related prompts during training and inference, demonstrating remarkably superior performance.

Prompt-based CL methods typically involve creating a set of learnable prompts, which are then optimized throughout the sequential learning of tasks. The core idea of these methods is to establish a unified query mechanism during training and inference, where the most relevant prompt to the current input are identified and incorporated into the pretrained model to generate instructed features for prediction. As shown in Figure 1, some methods Wang et al. (2022a;b) utilize a frozen vision transformer (ViT) to obtain uninstructed features and select prompts based on the cosine similarity between these features and learnable keys, which are associated with the prompts. The matching result is implicitly treated as inferred task identity (in gray). While the newly proposed HiDe-Prompt Wang et al. (2024a) trains a separate model to predict labels, explicitly incorporating task

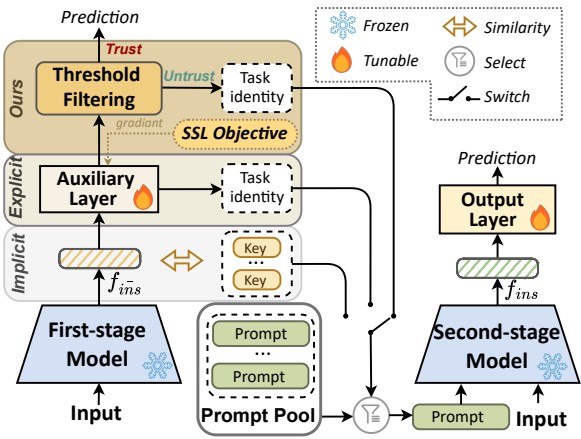

Figure 1: Illustration of prompt-based CL methods.

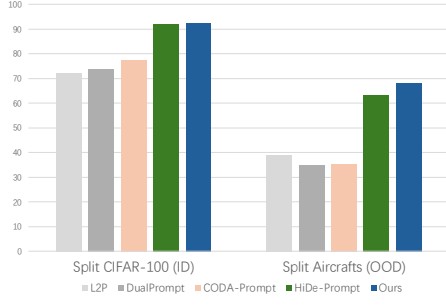

Figure 2: Performance comparison of different prompt-based CL methods on Split CIFAR-100 (ID) and Split Aircrafts (OOD) using iBOT model pretrained on ImageNet-21K, all datasets were splitted into ten incremental tasks.

identity information through a label-to-task mapping to choose prompts (in beige).[1] It significantly bridges the performance gap between self-supervised and supervised pretrained models. However, our empirical analysis reveals that these methods still fall short under out-of-distribution (OOD) scenarios, especially in comparison to their remarkably high performance under in-distribution (ID) scenarios, as shown in Figure 2.

In this paper, we first conducted a series of empirical analyses to thoroughly investigate the behavior of the existing prompt-based CL methods across two different scenarios (i.e. ID and OOD). The main observations are as follows: (1) self-supervised and supervised pretrained models perform similarly in ID scenarios, but self-supervised models excel in leveraging task identity in OOD settings. (2) A reliable task identity, which is model-agnostic, benefits the second stage without requiring identical architectures for both stages. Based on these observations, we first incorporated self-distillation loss while training first-stage model to enhance the accuracy of its predicted task identities. During inference, we found not all data requires two-stage inference, especially when the first-stage model has already provided reliable predictions. We first modeled the prediction confidence of the first-stage model using a $\beta$ distribution, and then proposed a simple yet effective threshold filtering strategy. This approach mitigates issues encountered under OOD scenarios, where second-stage model predicts incorrectly even with the correct task identity. Concretely, we chose the lowest boundary of a specific highest density interval of the distribution as the threshold for making decisions. We use this threshold to selectively send data to the second-stage model. Ultimately, we explored leveraging self-supervised pretrained models to improve the accuracy of task identity inference, thereby enhancing the models' continual learning capabilities.

The main contributions of this paper are summarized as follows: (1) We revisited existing prompt-based CL methods, analyzing the performance differences these methods exhibit across different scenarios and their main limitations. (2) We improved the first-stage model's adaptability with self-supervised learning, implemented a threshold filtering strategy to reduce second-stage errors from reliable predictions, and used self-supervised pretrained models as a unified task identifier, eliminating the need for identical architectures. (3) We achieved substantial improvements over state-of-the-art methods in OOD benchmarks and obtained comparable results in ID benchmarks.

## 2 RELATED WORK

**Continual Learning.** Continual learning has received increasing attention from researchers. Early approaches Zhu et al. (2021); Yu et al. (2020; 2022); Liu et al. (2022); Tao et al. (2024) tended to train a deep neural network from scratch, broadly divided into three categories. The first is replay-based methods, which alleviate forgetting by replaying stored previous exemplars Rebuffi et al. (2017); Hou et al. (2019) or generated pseudo samples Shin et al. (2017); Ostapenko et al. (2019).

---

[1]For clarity, we refer to the model inferring task identity as the "first-stage model" and the model that uses selected prompts to generate instructed features as the "second-stage model." The following text will follow this terminology.

The second is regularization-based methods, which typically apply constraints to the model Li & Hoiem (2017) by combining knowledge distillation Hinton (2015); Yu et al. (2019) techniques or restricting changes of important parameters Kirkpatrick et al. (2017). The third is architecture-based methods, which either expand the model when learning new tasks Zhu et al. (2022); Zhou et al. (2022) or assign different parts of the model's parameters to different tasks through masking Serra et al. (2018). Some methods Douillard et al. (2022); Zhai et al. (2023a; 2024) employ a combination of these techniques to achieve superior performance.

Approaches equipped with large-scale pretrained models (PTMs) have shifted the traditional CL paradigms. By extracting robust features from PTMs, some works directly take advantage of this to construct classifiers based on PTMs. SLCA Zhang et al. (2023) applied varying learning rates for representation layer and classifier to mitigate progressive overfitting, and reduce classification bias through prototype replay. RanPAC McDonnell et al. (2024) employed a frozen random projection layer to project pretrained features into a higher-dimensional space for better linear separability. Some works applied mixture of experts (MoE) methods, LAE Gao et al. (2023) employed an on-line module for learning new tasks and an offline module for preserving learned knowledge, with final predictions during the inference stage derived from the maximum logit of the two modules. ESN Wang et al. (2023) trained a separate classifier for each new task and introduced an anchor-based energy self-normalization strategy, with a voting-based strategy during inference stage to unify the classifiers. The recent proposed Yu et al. (2024) expanded the vision-language model Radford et al. (2021) through MoE and designed a distribution discriminator to dynamically allocate test samples to either MoE adapters or the original CLIP during inference.

**Prompt-based CL Methods.** Such approaches typically create a set of learnable prompts and predict by incorporating the prompts most relevant to the current input. L2P Wang et al. (2022b) selects prompts based on the cosine similarity between pretrained features and learnable keys, and integrates the selected prompts into the token sequence after the image is patchified by ViT, before feeding it to the transformer encoder. DualPrompt Wang et al. (2022a) categorizes prompts into general prompts, which are shared by all samples, and expert prompts that are attached to the key and value following L2P. CODA-Prompt Smith et al. (2023) proposed learning a set of prompt components to generate attention scores for weighting the prompts. HiDe-Prompt Wang et al. (2024a) adds a learnable MLP to the frozen ViT and uses prototype replay for sequential training first-stage model to directly predict class labels. It retrieves task identities by mapping them with class labels, allowing for the explicit selection of prompts based on the predicted task identities to obtain instructed features.

## 3 PRELIMINARY

Prompt-based CL methods typically create a set of learnable prompts to generate instructed features. In the case of vision tasks, for input image $x \in \mathbb{R}^{H \times W \times C}$, a pretrained ViT $f_\theta$ first divides it into $N$ non-overlapping patches. It then attaches a class token and incorporated position encodings into these patch embeddings to form a token sequence. This sequence is then processed through $L$-layers of stacked multi-head self-attention (MSA) blocks. We denote the input sequence of the $i$-th MSA layer as $x_e^i \in \mathbb{R}^{(N+1) \times D}$. For the $i$-th MSA layer, Query $x_q^i$, Key $x_k^i$, and Value $x_v^i$ are first created by multiplying the identical input sequence $x_e^i$ with projection matrices $W_Q^i$, $W_K^i$, and $W_V^i$, respectively. Then it calculates the self-attention scores and produces the output sequence through the projection matrix $W_O^i$ as:

$$x_j = Attention(x_q^i W_Q^{i,j}, x_k^i W_K^{i,j}, x_v^i W_V^{i,j}), j = 1, 2, ..., m \tag{1}$$

$$x_e^{i+1} = MSA(x_q^i, x_k^i, x_v^i) = Concat(x_1, ..., x_m)W_O^i \tag{2}$$

where $Attention(Q, K, V) = \frac{QK^T}{\sqrt{D}}V$, $m$ is the number of attention heads, $x_j$ is the output of the $j$th head.

Existing prompt-based CL methods generally adopt two techniques: prompt tuning Lester et al. (2021) and prefix tuning Li & Liang (2021). Prompt tuning appends learnable tokens to the sequence $x_e^i$ before it proceeds to the next MSA layer, whereas prefix tuning involves appending them to the $x_k^i$ and $x_v^i$ sequences.

Table 1: Performance of supervised and self-supervised pretrained ViT models on Split CIFAR-100 and Split Aircrafts datasets, both datasets were splitted into ten incremental tasks.

| | Split CIFAR-100 (In-distribution) | | | Split Aircrafts (Out-of-distribution) | | |
|---|---|---|---|---|---|---|
| | **Task ID** | **Final Label** | **Proportion(%)** | **Task ID** | **Final Label** | **Proportion(%)** |
| ViT-B-16 supervised pretrained on ImageNet-21K | ✓ | ✓ | 78.68 | ✓ | ✓ | 27.57 |
| | ✓ | ✗ | 1.89 | ✓ | ✗ | 12.90 |
| | ✗ | ✓ | 14.28 | ✗ | ✓ | 17.49 |
| | ✗ | ✗ | 5.15 | ✗ | ✗ | 42.04 |
| | **Task ID** | **Final Label** | **Proportion(%)** | **Task ID** | **Final Label** | **Proportion(%)** |
| ViT-B-16-DINO self-supervised pretrained on ImageNet-1K | ✓ | ✓ | 79.42 | ✓ | ✓ | 49.27 |
| | ✓ | ✗ | 2.51 | ✓ | ✗ | 9.93 |
| | ✗ | ✓ | 10.68 | ✗ | ✓ | 15.27 |
| | ✗ | ✗ | 7.39 | ✗ | ✗ | 25.08 |

Table 2: Various combinations and effects between the first-stage model and second-stage model, experiments were conducted on Split Aircrafts, splitted into ten incremental tasks.

| First-stage model | Second-stage model | Task accuracy | Final accuracy |
|---|---|---|---|
| ViT-B-16-IN21K | ViT-B-16-IN21K | 40.47 | 45.06 |
| ViT-B-14-DINOv2 | ViT-B-16-IN21K | 73.39 | 58.18 |

As illustrated in Figure 1, some methods Wang et al. (2022a;b); Smith et al. (2023) calculate the cosine similarity between uninstructed features and keys, implicitly treating the matching results as inferred task identity information (in gray). In contrast, a recent state-of-the-art method, HiDe-Prompt Wang et al. (2024b), trains a model and explicitly uses its prediction as task identity to select prompts (in beige). For prompt-based CL methods, an accurate task identity helps integrate relevant information for correct predictions, while an incorrect identity typically leads to misclassification by incorporating wrong information. To address the limitations of current approaches, we propose a novel framework (in brown) that incorporates self-distillation loss and a threshold filtering strategy.

## 4 EMPIRICAL ANALYSIS

We will investigate two key questions regarding prompt-based CL methods: *1. How do the supervised pretrained and self-supervised models perform under ID and OOD scenarios? 2. Does the first-stage model, which infers task identity, need to be identical to the second-stage model, which incorporates prompts and provides the final predictions?* Our experiments are conducted using the HiDe-Prompt method Wang et al. (2024a), which has shown impressive performance across various benchmarks. We will present our observations based on these findings.

**Self-supervised models excel under OOD scenarios.** To demonstrate the performance of prompt-based CL methods across different distribution scenarios, we conducted a comparison of two ViT-B-16 models: one supervised pretrained on ImageNet-21K and the other self-supervised pretrained on ImageNet-1K using DINO Caron et al. (2021), across in-distribution dataset Split CIFAR-100 and out-of-distribution dataset Split Aircrafts. The results shown in Table 1 indicate that both models effectively utilized task identity under ID scenario, rarely making errors when the task identity is correct, with incorrect predictions on only 1.89% and 2.51% of test samples under this condition. The proportion of correctly predicted samples was as high as 78.68% and 79.42%. Under OOD scenarios, these two models exhibit significantly different performance. Specifically, the self-supervised model was much better at leveraging the task identity, with 21.7% higher than supervised pretrained model when both the task identity and the final prediction were correct.

*Observation 1: Under ID scenarios, both self-supervised and supervised pretrained models perform similarly, rarely making errors. However, under OOD scenarios, self-supervised models significantly outperform supervised ones in leveraging task identity.*

**Task identity provision is model-agnostic.** Previous methods typically restricted the first-stage model and second-stage model to use the same architecture and pretrained weights. The accuracy of the task identity provided by the first-stage model is critical for the subsequent task-specific

prompt selection by the second-stage model. If the first-stage task predictions are unreliable, it will inevitably affect the final performance outputed from the second-stage model. The results presented in Table 2 for the Aircraft dataset indicate that when using the same model, ViT-B-16-IN21K, for both the first and second stages, the final accuracy reaches 45.06%. we then replaced the first-stage model with ViT-B-14-DINOv2, which significantly improved task accuracy by 32.92%, leading to an overall gain of 13.12% in final accuracy.

**Observation 2**: *Providing task identity is a model-agnostic behavior, a reliable task identity benefits the second stage without requiring identical architecture for both stages.*

## 5 METHOD

### 5.1 BOOSTING CLASS INCREMENTAL LEARNING VIA SELF-SUPERVISED LEARNING

Our empirical analysis in Section 4 suggests that the first-stage and second-stage models can be trained using different architectures. We propose incorporating self-supervised learning into our framework from two angles: utilizing self-supervised pretrained models and applying self-supervised loss to enhance task identity prediction.

#### 5.1.1 WITH SELF-SUPERVISED PRETRAINED MODEL

Supervised pretrain often leads to neural collapse Galanti et al. (2021); Papyan et al. (2020); Fang et al. (2021); Zhai et al. (2023b), where features of the same class cluster around their mean, making it hard to generalize to OOD scenarios where more fine-grained discriminative ability is required. Self-supervised pretrain avoids this issue by learning general representations capable of generalizing to novel or unseen data, thereby enhancing OOD performance, which is especially advantageous for CL. Considering the high transferability of features in self-supervised pretrained models and their inherent advantages under OOD scenarios, using them as the first-stage models to provide task identity proves more effective. Specifically, in this work, we explore models that have been self-supervised pretrained at different scales with various pretrain methods, including iBOT Zhou et al. (2021) pretrained on ImageNet-21K/1K, DINO Caron et al. (2021) and MoCo v3 Chen et al. (2021) pre-trained on ImageNet-1K, and DINOv2 Oquab et al. (2023) pre-trained on LVD-142M.

#### 5.1.2 WITH SELF-SUPERVISED LOSS

We incorporated self-distillation loss into our approach, a self-supervised learning objective derived from the DINO (self-distillation with no labels) framework Caron et al. (2021), which is effective in enhancing feature extraction capabilities of vision transformers. During training the first-stage model, we consider the current network as $f_{\theta_s}$ and replicate it as the teacher network, denoted as $f_{\theta_t}$, parameterized by $\theta_s$ and $\theta_t$ respectively. Initially, we generate a set of augmented views $V$ from the input image $x$, including two global views $\{x_1^g, x_2^g\}$ and several local views. Then all views are fed into the student network, while the global views are input only into the teacher network. Cross-entropy loss is minimized between the outputs of the two networks to match their distributions:

$$\min_{\theta_s} \sum_{x \in \left\{x_g^1, x_g^2\right\}} \sum_{\substack{x' \in V \\ x' \neq x}} H(P_t(x), P_s(x')) \tag{3}$$

Where $H(a, b) = -a \log b$, $P(x)$ represents the network's probability distribution over $K$ dimensions, calculated by the following equation:

$$P(x)^{(i)} = \frac{\exp(f_\theta(x)^{(i)}/\tau)}{\sum_{k=1}^{K} \exp(f_\theta(x)^{(k)}/\tau)} \tag{4}$$

with $f_{\theta_s}$, $\tau_s$ for student network, and $f_{\theta_t}$, $\tau_t$ for teacher network. The parameters of the student network $\theta_s$ are optimized via stochastic gradient descent during the minimization of 3, while the parameters of the teacher network $\theta_t$ are updated through the EMA (Exponential Moving Average) algorithm Morales-Brotons et al. (2024).

Applying self-distillation loss (SDL) for CL on downstream data streams offers several advantages. Firstly, the computation of self-distillation loss effectively acts like traditional knowledge

distillation-based regularization methods for preventing forgetting, but it relaxes constraints to maintain model plasticity. By promoting consistency between multiple views (global and local) of the same input, it facilitates the extraction of generalized features from images. The learned features are task-invariant, making these generalized features favorable for the model's improved understanding and adaptation to new and unseen data, which is crucial for CL. Secondly, by updating the teacher network with EMA, which is more stable and updates slowly, old knowledge is preserved. By continuously distilling knowledge from the teacher to the student and utilizing different views of the data, SDL can potentially mitigate catastrophic forgetting.

## 5.2 NOT ALL DATA NEED TWO-STAGE INFERENCE

Our analysis in Section 4 reveals a key issue with the existing prompt-based CL methods: even if the first-stage model correctly predicts the class label, the second-stage model may still misclassify the category, despite receiving the correct task identity. This situation occurs frequently in OOD scenarios, with an occurrence rate of around 10%. To mitigate this issue, we propose a simple yet effective threshold filtering strategy to determine which samples require two-stage inference and which do not. We utilize $\beta$ distribution for modeling confidence scores and choosing the lowest boundary of a specific highest density interval as the decision threshold. Predictions from the first-stage model are used directly for samples above this threshold, while samples below this threshold are fed to the second-stage model. Specially, given an input image $x$, the output probability for class $i$ of this image is $P_i = \frac{e^{z_i}}{\sum_j e^{z_j}}$, where $z_i$ is its logits, $j$ is the number of classes encountered so far. The predicted class $\hat{y} = arg \max_i P_i$ is the class with the highest probability. The confidence $c = \max_i P_i$ that the model assigns to $x$ is the probability of the predicted class.

Assume that the confidence score follows $\beta$ distribution, then the prior distribution can be represented as: $\theta \sim Beta(\alpha_0, \beta_0)$, where $\alpha_0$ and $\beta_0$ are two parameters of the $\beta$ distribution, which control the shape of the distribution.

For $n$ observed confidence scores from all test samples so far $c_1, c_2, ..., c_n$, assuming they are also sampled from $\beta$ distribution, the likelihood function is:

$$L(\theta) = \prod_{i=1}^{n} c_i^{\alpha-1} \cdot (1 - c_i)^{\beta-1} \tag{5}$$

According to Bayes theorem, combining the prior distribution and the likelihood function, the posterior distribution is:

$$P(\theta|c_1, \ldots, c_n) \propto \left( \prod_{i=1}^{n} c_i^{\alpha-1} \cdot (1 - c_i)^{\beta-1} \right) \cdot \text{Beta}(\alpha_0, \beta_0) \tag{6}$$

Since both the prior and likelihood functions are $\beta$ distributions, the posterior distribution is still $\beta$ distribution, and the updated parameters are:

$$\alpha_{post} = \alpha_0 + \sum_{i=1}^{n} c_i, \beta_{post} = \beta_0 + n - \sum_{i=1}^{n} c_i \tag{7}$$

$$\theta_{post} \sim Beta(\alpha_{post}, \beta_{post}) \tag{8}$$

After obtaining the posterior distribution, the confidence threshold can be determined by calculating the quantile of the posterior distribution. For example, the lower limit of the $d\%$ highest density interval (HDI) is selected as the confidence threshold for classification:

$$\tau = Beta^{-1}(d|\alpha_{post}, \beta_{post}) \tag{9}$$

where $Beta^{-1}$ represents the quantile function (or inverse cumulative distribution function) of the $\beta$ distribution.

For samples that exceed the threshold $\tau$, we consider that the predictions from the first-stage model are reliable and adopt them. For samples below $\tau$, we fed them to the second-stage model for a secondary inference phase, guided by the inferred task identity.

Table 3: Performance comparison of various methods on Split Aircrafts and Split Cars-196 Datasets, we present FAA, CAA, and FFM, each metric with mean and standard deviation over three different random seeds. The best outcome is marked in bold, with the second-best underlined. All experimental results for compared methods were reproduced by ourselves.

| PTM (A-B) | Method | Split Aircrafts | | | Split Cars-196 | | |
|---|---|---|---|---|---|---|---|
| | | **FAA** (↑) | CAA (↑) | FFM (↓) | **FAA** (↑) | CAA (↑) | FFM (↓) |
| Sup-21K* | L2P | 22.76 ±0.66 | 35.99 ±1.18 | 21.22 ±3.79 | 34.49 ±0.19 | 45.97 ±1.40 | 12.39 ±2.18 |
| | DualPrompt | 23.82 ±1.76 | 35.88 ±1.85 | 16.76 ±1.87 | 43.21 ±0.50 | 51.72 ±1.46 | 12.00 ±1.76 |
| | CODA-Prompt | 19.02 ±0.99 | 35.69 ±1.14 | 34.04 ±4.26 | 33.12 ±0.13 | 50.16 ±1.05 | 34.54 ±2.19 |
| | HiDe-Prompt | 44.21 ±0.79 | 52.75 ±1.78 | 10.04 ±0.54 | 49.75 ±0.18 | 58.47 ±0.87 | 7.80 ±0.22 |
| | **Ours** | **47.47 ±0.34** | **55.45 ±1.75** | **7.77 ±0.44** | **54.90 ±0.10** | **62.23 ±1.15** | **7.59 ±0.39** |
| iBOT-21K | L2P | 33.26 ±2.82 | 50.62 ±2.37 | 15.74 ±2.28 | 49.42 ±0.86 | 61.02 ±1.08 | 10.29 ±0.46 |
| | DualPrompt | 30.77 ±2.59 | 46.28 ±2.25 | 23.21 ±4.31 | 46.91 ±0.80 | 57.92 ±1.17 | 13.25 ±0.61 |
| | CODA-Prompt | 36.52 ±1.04 | 53.43 ±1.44 | 21.05 ±2.72 | 60.28 ±0.43 | 70.10 ±0.18 | 11.89 ±0.43 |
| | HiDe-Prompt | 60.24 ±1.52 | 63.44 ±3.70 | **4.45 ±0.16** | 68.23 ±0.50 | 71.67 ±1.37 | **3.07 ±0.14** |
| | **Ours** | **66.58 ±0.77** | **71.36 ±1.59** | 5.15 ±0.52 | **71.51 ±0.30** | **75.15 ±0.80** | 5.03 ±0.13 |
| iBOT-1K | L2P | 34.82 ±2.33 | 51.06 ±1.87 | 17.55 ±1.18 | 52.40 ±0.53 | 62.90 ±1.20 | 10.33 ±0.99 |
| | DualPrompt | 34.25 ±1.79 | 49.27 ±2.37 | 21.67 ±2.70 | 53.10 ±0.62 | 64.75 ±0.84 | 14.88 ±0.54 |
| | CODA-Prompt | 39.24 ±0.60 | 56.10 ±1.65 | 18.56 ±2.32 | 62.11 ±1.09 | 72.48 ±0.77 | 10.74 ±0.86 |
| | HiDe-Prompt | 61.37 ±1.68 | 64.14 ±3.57 | **4.59 ±0.54** | 70.67 ±0.11 | 73.91 ±1.16 | **3.95 ±0.25** |
| | **Ours** | **65.54 ±0.52** | **70.34 ±1.55** | 5.82 ±0.22 | **74.94 ±0.04** | **78.58 ±0.76** | 4.87 ±0.52 |
| DINO-1K | L2P | 36.52 ±0.97 | 50.08 ±1.42 | 17.85 ±3.34 | 51.68 ±0.69 | 62.36 ±0.63 | 11.02 ±1.97 |
| | DualPrompt | 36.05 ±2.46 | 52.15 ±2.81 | 18.46 ±2.68 | 52.97 ±0.83 | 64.21 ±1.25 | 11.47 ±1.32 |
| | CODA-Prompt | 42.71 ±1.42 | 57.92 ±0.82 | 17.66 ±0.93 | 62.43 ±0.74 | 72.13 ±0.51 | 10.29 ±0.24 |
| | HiDe-Prompt | 62.14 ±1.44 | 65.55 ±3.55 | **4.49 ±0.49** | 71.93 ±0.08 | 75.57 ±0.91 | **3.82 ±0.22** |
| | **Ours** | **66.12 ±0.63** | **71.11 ±1.81** | 5.88 ±0.34 | **75.39 ±0.11** | **78.95 ±0.79** | 4.61 ±0.17 |
| MoCo-1K | L2P | 26.84 ±1.07 | 43.69 ±2.25 | 9.08 ±1.47 | 39.10 ±0.39 | 54.22 ±0.81 | **3.22 ±0.15** |
| | DualPrompt | 27.68 ±2.01 | 43.55 ±2.03 | 9.58 ±2.90 | 41.73 ±0.80 | 56.68 ±1.00 | 3.29 ±0.38 |
| | CODA-Prompt | 35.75 ±1.56 | 52.11 ±1.90 | 20.12 ±2.78 | 54.06 ±0.21 | 65.25 ±1.17 | 16.57 ±0.51 |
| | HiDe-Prompt | 53.05 ±0.89 | 58.74 ±2.79 | **4.89 ±0.20** | 66.09 ±0.36 | 70.75 ±1.30 | 3.87 ±0.11 |
| | **Ours** | **57.52 ±0.52** | **63.75 ±1.80** | 7.12 ±0.29 | **68.64 ±0.14** | **72.92 ±1.11** | 5.03 ±0.45 |

# 6 EXPERIMENTS

## 6.1 EXPERIMENTAL SETUP

**Datasets.** For evaluating the CIL performance of different methods under OOD scenarios, we chose Split Aircrafts Maji et al. (2013) and Split Cars-196 Krause et al. (2013) datasets. Additionally, we also conducted experiments on CIFAR-100 Krizhevsky et al. (2009), ImageNet-R Hendrycks et al. (2021), and CUB-200 Wah et al. (2011); the first two are subclasses of ImageNet, whereas CUB-200 has 52 overlapping categories as ImageNet Ostapenko et al. (2022); Wen et al. (2022). Cars-196 was divided into 7 tasks, all other datasets were divided into 10 tasks each. Detailed descriptions of these datasets are available in the appendix.

**Implementation Details.** Following Wang et al. (2024a), we used ViT-Base model with a patch size of 16 (except for DINOv2 Oquab et al. (2023), which is 14), set the projection dimension to 2048 for computing SDL, and used d=95 to determine the threshold. The weight of SDL was set to 0.1, with all other training hyperparameters consistent with those in Wang et al. (2024a). Following the evaluation metrics in Wang et al. (2024a), we reported the final average accuracy (FAA) of all seen classes, the average accuracy over tasks refereed as cumulative average accuracy (CAA), and final forgetting measure (FFM) of all previous tasks. The models are either supervised pretrained Ridnik et al. (2021) or self-supervised pretrained Zhou et al. (2021); Caron et al. (2021); Chen et al. (2021) on ImageNet-21K/1K, aligning with the benchmarks for a fair comparison. In extended experiments, we also utilized DINOv2 Oquab et al. (2023) pretrained on LVD-142M.

Table 4: FAA(↑) comparison for various methods on three different datasets.

| Method | Split ImageNet-R | | | | | Split CUB-200 | | | | |
|---|---|---|---|---|---|---|---|---|---|---|
| | Sup-21K | iBOT-21K | iBOT-1K | DINO-1K | MoCo-1K | Sup-21K | iBOT-21K | iBOT-1K | DINO-1K | MoCo-1K |
| L2P | 59.61 | 61.23 | 64.16 | 61.01 | 55.01 | 73.92 | 52.57 | 57.55 | 53.79 | 52.94 |
| DualPrompt | 65.83 | 60.50 | 64.81 | 60.75 | 54.06 | 74.55 | 48.09 | 56.62 | 55.75 | 52.04 |
| CODA-Prompt | 59.93 | 66.72 | 69.00 | 63.60 | 61.50 | 72.25 | 54.73 | 59.96 | 60.01 | 45.79 |
| HiDe-Prompt | **73.26** | 74.73 | **76.82** | **73.90** | 67.58 | 83.26 | 70.73 | 77.24 | **76.75** | 71.82 |
| Ours | 72.20 | **75.07** | 76.29 | 73.58 | **68.55** | **83.59** | **71.92** | **77.94** | 76.53 | **72.59** |

| Method | Split CIFAR-100 | | | | |
|---|---|---|---|---|---|
| | Sup-21K | iBOT-21K | iBOT-1K | DINO-1K | MoCo-1K |
| L2P | 83.43 | 72.09 | 75.59 | 79.79 | 77.32 |
| DualPrompt | 87.98 | 73.65 | 77.49 | 78.10 | 73.80 |
| CODA-Prompt | 81.30 | 77.25 | 78.79 | 81.48 | 79.56 |
| HiDe-Prompt | **92.96** | 91.92 | 92.04 | 90.10 | **90.57** |
| Ours | 92.35 | **92.49** | **92.38** | **90.86** | 90.30 |

## 6.2 COMPARISON TO STATE-OF-THE-ART METHODS

**Results on OOD scenarios.** Table 3 displays the overall results of our method compared to four popular prompt-based CL methods as we mentioned above on two OOD datastets: the Split Aircrafts and Split Cars-196. For all approaches[2], self-supervised pretrained models (without ∗) demonstrate markedly better performance than supervised pretrained models (with ∗), consistent with our observations that self-supervised pretrained models have significant advantages under OOD scenarios. Among all the methods, the proposed method demonstrated significant improvement over all four compared methods across all models For supervised pretrained models, we achieved an FAA increase of 3.26% and a CAA increase of 2.70% on Split Aircrafts, with improvements of 5.15% and 3.76% respectively on Split Cars-196, while also achieving the lowest forgetting on both datasets. Similar improvements were observed for self-supervised models, notably a 6.34% FAA and 7.92% CAA boost for the iBOT model pretrained on ImageNet-21K. The improvements in CAA highlight our method's consistent performance improvements across all stages of incremental learning, rather than only after all tasks have been trained. Additionally, our method exhibits the lowest standard deviation across almost all metrics, further indicating the stability of our approach. Compared to HiDe-Prompt, our method exhibited slightly higher forgetting on self-supervised models.

**Results on non-OOD scenarios.** We also evaluated our proposed method on Split ImageNet-R, CIFAR-100, and CUB-200, which are commonly assessed in previous methods. The overall results are shown in Table 4. We achieved improvements of 0.55% and 0.16% over the SOTA method across five different models on two ID datasets Split CUB-200 and Split CIFAR-100, respectively. For Split ImageNet-R, we observed certain improvements on two self-supervised pretrained models iBOT-21K and MoCo-1K, while experiencing slight declines on other models. In summary, although existing methods already perform excellently under ID scenarios, our method can still achieve certain improvements. However, as we previously mentioned, our method shows even greater advantages under OOD scenarios.

## 6.3 FURTHER ANALYSIS

**Impact of pretrain data scale.** In analyzing the effect of the scale of pretrain data, results in Table 5 present the final average accuracy (FAA) provided by the first-stage model and second-stage model across different data scales. We found that, in the first stage, supervised pretrained model significantly underperforming all self-supervised models on Split CIFAR-100 and falling far behind on Split Aircrafts, its strength lies in the second stage under ID scenarios (Split CIFAR-100), where it achieves a 12.94% improvement over the first stage—the highest among all models. For self-supervised pretrained models, except for MoCo-1K on Split Aircrafts, other models pretrained on ImageNet-1K exhibit comparable performance in both stages. With increased scale of pretrain data, the first-stage performance improves notably, while the improvement from the second stage over the first stage lessens. For DINOv2 with the largest scale of pretrain data, the second stage's

---

[2]We denote the supervised pretrained model as ∗, where 'Sup' stands for 'supervised' and '21K' refers to the ImageNet-21K dataset. The other four models are self-supervised, labeled in the format 'A-B', with A representing the pretrain method and B the dataset.

Table 5: Performance comparison across models on Split CIFAR-100 and Split Aircrafts datasets.

| | Pretrain method | Pretrain data | Split CIFAR-100 | | Split Aircrafts | |
|---|---|---|---|---|---|---|
| | | | first-stage FAA | second-stage FAA | first-stage FAA | second-stage FAA |
| Supervised | - | ImageNet-21K | 79.41 | 92.35 | 42.42 | 47.37 |
| Self-supervised | MoCoV3 | ImageNet-1K | 81.33 | 90.30 | 53.23 | 58.18 |
| | DINO | | 82.28 | 90.86 | 61.12 | 67.36 |
| | iBOT | | 83.39 | 92.38 | 60.72 | 66.49 |
| | iBOT | ImageNet-21K | 85.18 | 92.49 | 63.37 | 68.08 |
| | DINOv2 | LVD142M | 88.67 | 92.51 | 73.39 | 75.46 |

Table 6: The performance on Split CIFAR-100 and Aircrafts of different methods employing diverse combinations of the first-stage model and second-stage model, compared to our proposed method.

| Dataset | First-stage model | Second-stage model | L2P | Dual | HiDe | Ours |
|---|---|---|---|---|---|---|
| Split CIFAR-100 | Sup-21K | Sup-21K | 83.43 | 87.98 | **92.96** | 92.35 |
| | DINO-1K | | 81.83 | 87.77 | **93.01** | 92.53 |
| | DINOv2-LVD142M | | 82.15 | 88.09 | 93.88 | **94.18** |
| Split Aircrafts | Sup-21K | Sup-21K | 24.07 | 24.75 | 45.06 | **47.37** |
| | DINO-1K | | 25.26 | 28.30 | 53.29 | **61.06** |
| | DINOv2-LVD142M | | 25.56 | 29.43 | 58.18 | **70.01** |

improvement over the first on Split CIFAR-100 is only 3.84%, and the relative improvement on Split Aircrafts is merely 2.07% (for comparison, although not shown, it is only 0.33% on HiDe-Prompt).

**Combination of different models.** We explored different combinations of first-stage and second-stage models on Split CIFAR-100 and Split Aircrafts datasets, with results shown in Table 6. Existing methods that often use identical architectures and weights for both models are outlined in the first row. For Split Aircrafts, by replacing the first-stage model with self-supervised pretrained models (DINO-1K and DINOv2-LVD142M), we observed significant improvements both in explicit task identity prediction methods (HiDe-Prompt and ours) and in implicit task identity usage methods like L2P and DualPrompt, which rely on matching results between pretrained features and learnable keys. This indicates that supervised pretrained models might lack sufficient capability to encode more fine-grained features under OOD scenarios, highlighting the necessity of employing self-supervised pretrained models. For our method, we achieved substantial improvements by enhancing the accuracy of task identity and filtering first-stage predictions. On Split CIFAR-100, previous methods that implicitly infer task identity experienced a slight decrease when the first-stage model was replaced with a self-supervised model, except for a modest increase when using DINOv2 with DualPrompt. However, explicit methods continue to show significant enhancements, particularly when applying DINOv2 to our method. Combining the high first-stage accuracy of DINOv2 with the extremely high efficacy of the supervised pretrained model in the second stage, we achieved an impressive accuracy of 94.18%. We show more results from different model combinations in Table 9, further analysis can be found in the appendix.

**Class similarity for different models.** Following the metric proposed in Ostapenko et al. (2022), we assess the discriminative capabilities of different models across various scenarios by utilizing class similarity $S$, defined as the average of pairwise similarities among the $c$ classes in a given dataset as follows:

$$S = \frac{1}{T} \sum_{i=1}^{c} \sum_{j=i+1}^{c} cos(p_i, p_j) \qquad (10)$$

Where $T$ is the number of possible pairs of classes, $cos$ is the cosine similarity function, $p_i$ is the mean vector of extracted features belong to class $i$. On one hand, $S$ reflects the difficulty of classification on that dataset; the lower the similarity, the more orthogonal the features, and consequently, the easier the classification. On the other hand, this metric can also be used to indicate the extent to which the pretrained model leaks information about downstream data.

The heatmaps of the similarity matrices between classes are shown in Figure 3. The specific values for $S$ are in Table 8, which can be found in the appendix. The results indicate that class similarity $S$

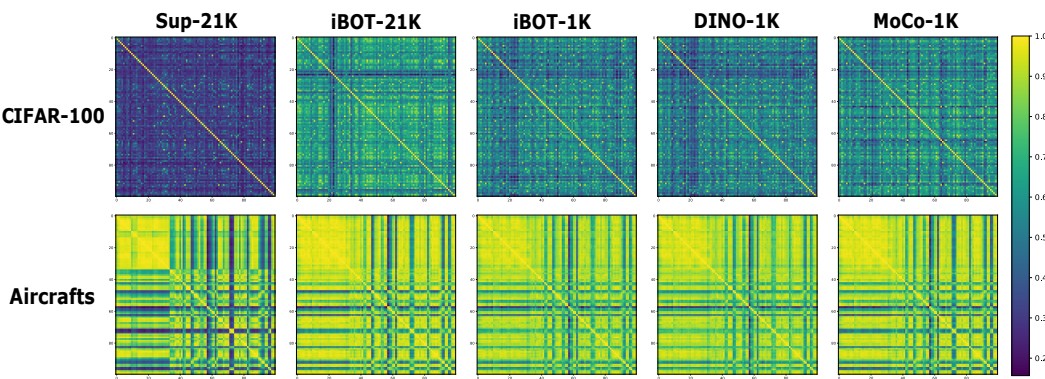

Figure 3: Class similarity heatmaps for different models on CIFAR-100 (top row) and Aircrafts (bottom row) datasets. Each heatmap represents the pairwise class similarity matrix, with the color scale indicating similarity values. The lower the feature similarity indicates more orthogonal features and lower classification difficulty. The higher suggests that the downstream dataset is more out-of-distribution compared to the pretrained data.

Table 7: Ablation studies of our proposed method, we present FAA(↑) for comparison.

| Method | Split Aircrafts | | | | | Split Cars-196 | | | | |
|---|---|---|---|---|---|---|---|---|---|---|
| | *Sup-21K* | *iBOT-21K* | *iBOT-1K* | *DINO-1K* | *MoCo-1K* | *Sup-21K* | *iBOT-21K* | *iBOT-1K* | *DINO-1K* | *MoCo-1K* |
| HiDe-Prompt | 45.06 | 63.19 | 64.72 | 64.99 | 54.76 | 49.47 | 68.86 | 70.80 | 71.86 | 66.56 |
| +SDL | 46.09 | 64.39 | 64.51 | 66.82 | 56.41 | 51.47 | 70.07 | 73.85 | 74.43 | 67.74 |
| +threshold | 47.37 | 68.08 | 66.49 | 67.36 | 58.18 | 54.72 | 72.08 | 74.88 | 75.49 | 68.83 |

on CIFAR-100 is relatively low across all models, suggesting more orthogonal features and easier classification. This is particularly obvious in Sup-21K, which shows a similarity as low as 0.228. As all categories of CIFAR-100 are included in ImageNet Wen et al. (2022), the low similarity observed in supervised pretrained models on this dataset is expected. On the contrast, these models exhibit relatively higher class similarity on Aircrafts, despite the category *"aircraft"* being included in ImageNet. Nonetheless, accurately classifying specific aircraft types within this fine-grained dataset remains challenging for the models.

## 6.4 Ablation Study

We conducted ablation studies to validate the effectiveness of each component of our proposed method, with the results shown in Table 7. Both modules consistently improved performance across different models. Specifically, on Split Aircrafts, SDL improved performance in all models except for a slight decrease in iBOT-1K, while our proposed threshold filtering strategy provided further enhancements. On Split Cars-196, both modules showed stable improvements across all models.

## 7 Conclusions

In this work, we revisited existed prompt-based CL methods through comprehensive analysis. Through empirical analysis, we revealed several limitations of existing methods. To overcome these limitations, we significantly boosted the performance of various pretrained models under OOD scenarios by introducing self-supervised learning objectives in the first stage and proposing a simple threshold filtering strategy. Moreover, we explored the efficiency of self-supervised pretrained models in providing task identities, thereby achieving further improvements and contributing to establishing a more unified framework for these approaches.

**Limitations.** Although our proposed method shows significant improvement under OOD scenarios, the gains under ID scenarios are relatively less. Additionally, even though we have overcome the limitations of previous methods—allowing the first-stage model in our framework to be any architecture (ViT, CNN, etc.), the second-stage model still relies on the ViT structure to integrate prompts and generate instructed features for prediction.

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

# A APPENDIX

## A.1 CL DATASETS

**Aircrafts** includes 100 different categories, each representing a specific type of aircraft. It contains a total of 6667 training samples and 3333 testing samples.

**Cars-196** comprises 196 distinct categories, with each category corresponding to a specific type of car. The dataset consists of 8144 training images and 8041 test images.

**CIFAR-100** comprises 100 categories, divided into 20 superclasses, each containing 5 fine-grained categories. Each category consists of 600 images of size 32x32, with 500 designated for the training set and 100 for the testing set.

**ImageNet-R** consists of 200 subclasses extracted from ImageNet, each containing challenging or stylistically recollected samples. The dataset includes a total of 30000 samples without a standardized split between training and testing sets. Typically, previous methods Smith et al. (2023); Wang et al. (2024a) assign 80% of the samples (24000) for training and the remaining 20% (6000) for testing.

**CUB-200-2011** comprises 200 distinct categories of birds, featuring 5994 training samples and 5794 testing samples.

## A.2 CLASS SIMILARITY OF DIFFERENT MODELS

Table 8: Average class similarity for different models on testsets of CIFAR-100 and Aircrafts.

|           | Sup-21K | iBOT-21K | iBOT-1K | DINO-1K | MoCo-1K |
|-----------|---------|----------|---------|---------|---------|
| CIFAR-100 | 0.228   | 0.702    | 0.649   | 0.655   | 0.775   |
| Aircrafts | 0.782   | 0.905    | 0.923   | 0.923   | 0.946   |

## A.3 MORE EXPERIMENTS FOR DIFFERENT COMBINATIONS OF TWO-STAGE MODELS

We display more experimental results of different combinations of first-stage and second-stage models in Table 9. Here, DINOv2-LVD142M is consistently used as the first-stage model, while various self-supervised pretrained models are employed for the second stage. Our approach achieves substantial improvements over SOTA methods under both ID and OOD scenarios. Specifically, on Split CIFAR-100 (ID scenario), we achieve an average improvement of 0.94% over the other four self-supervised models besides DINOv2-LVD142M, with all results outperforming the case where DINOv2-LVD142M is used in both stages. This breaks the limitation discussed in our analysis in Table 5 regarding the relatively limited second-stage improvement, highlighting the superiority of our method. For Split Aircrafts (OOD scenario), we achieve significant gains in all situations, with an average improvement of 6.01% across five models. Particularly, the final performance on iBOT-1k and iBOT-21k surpasses that of using DINOv2-LVD142M in both stages.

Table 9: The performance on Split CIFAR-100 and Aircrafts of different methods employing diverse combinations of the first-stage model and second-stage model, compared to our proposed method.

| dataset | first-stage model | second-stage model | L2P | Dual | HiDe | Ours |
|---------|-------------------|--------------------|-----|------|------|------|
| Split CIFAR-100 | DINOv2-LVD142M | DINOv2-LVD142M | 85.60 | 85.84 | **92.54** | 92.51 |
|         |                   | MoCo-1K   | 75.28 | 78.00 | 91.81 | **93.06** |
|         |                   | DINO-1K   | 72.45 | 73.20 | 91.69 | **92.77** |
|         |                   | iBOT-1K   | 75.92 | 74.21 | 92.92 | **93.68** |
|         |                   | iBOT-21K  | 78.68 | 76.76 | 93.13 | **93.80** |
| Split Aircrafts | DINOv2-LVD142M | DINOv2-LVD142M | 33.07 | 29.06 | 72.61 | **75.46** |
|         |                   | MoCo-1K   | 29.40 | 32.46 | 66.23 | **73.70** |
|         |                   | DINO-1K   | 36.84 | 38.67 | 69.29 | **75.95** |
|         |                   | iBOT-1K   | 38.29 | 39.54 | 70.55 | **75.62** |
|         |                   | iBOT-21K  | 37.21 | 35.44 | 68.18 | **76.16** |

## A.4 PREDICTION CONFIDENCE HISTOGRAMS OF DIFFERENT MODELS

Figure 4 illustrates the prediction confidence histograms of five different models on the test sets of CIFAR-100 and Aircrafts after the first stage of training. Red indicates incorrect classifications, and blue indicates correct classifications. The x-axis represents prediction confidence, and the y-axis represents the number of samples. It is evident that on CIFAR-100, all models exhibit very high confidence for correctly classified samples, whereas misclassified samples have generally lower confidence. Under OOD scenario with Split Aircrafts, supervised pretrained models show a significantly smaller number of correctly predicted samples with high confidence compared to self-supervised pretrained models. For incorrectly predicted samples, self-supervised pretrained models not only have generally lower confidence but also significantly fewer such samples.

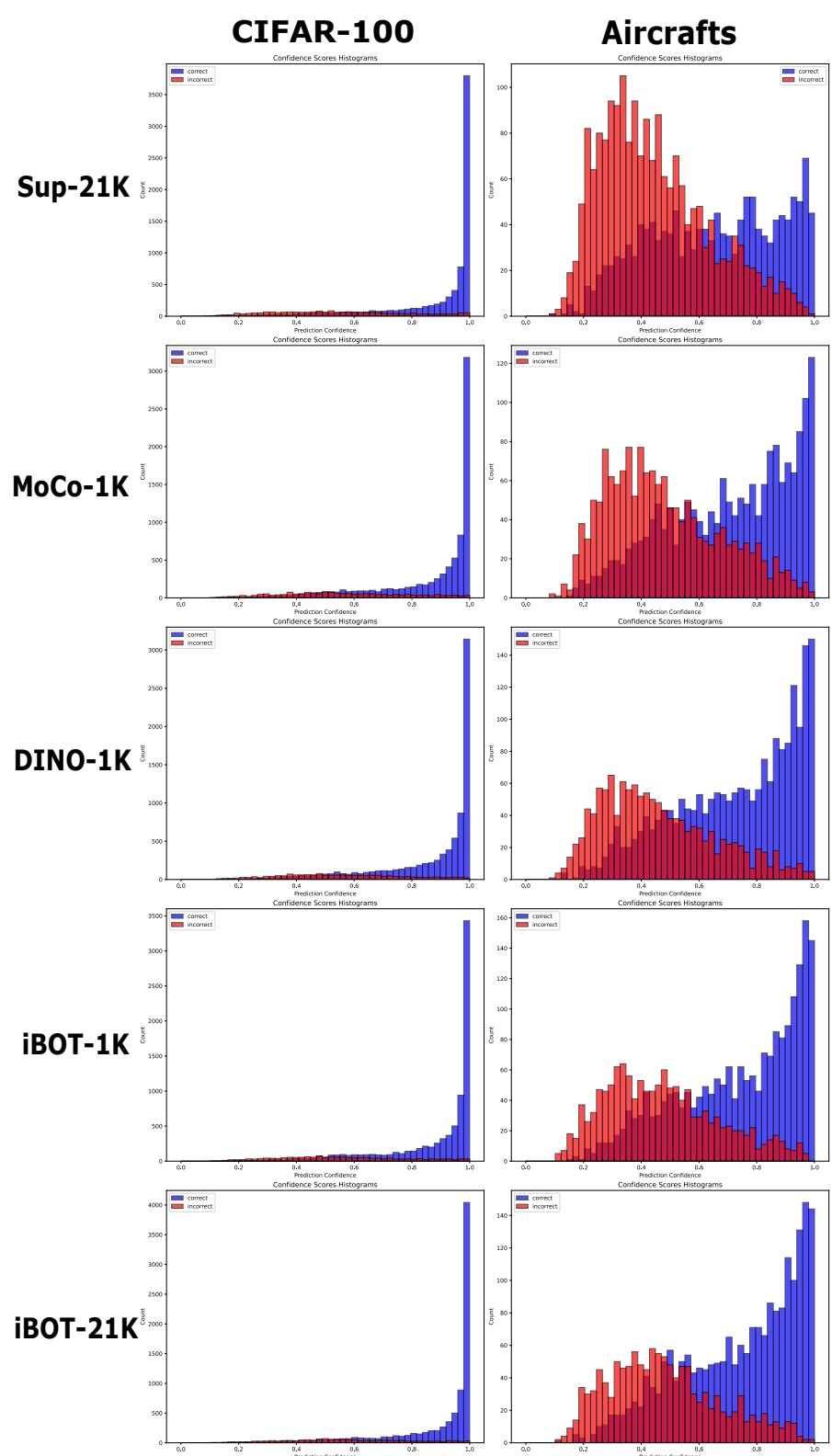

Figure 4: Confidence histograms on two datasets.

