# OpenReview forum: "Revisiting Prompt-based Methods in Class Incremental Learning"
_ICLR.cc/2025/Conference — ICLR 2025 Conference Withdrawn Submission_

### Official Review · Reviewer_Jn5R · 2024-10-17

**Soundness:** 2
**Presentation:** 1
**Contribution:** 2
**Rating:** 3
**Confidence:** 5

**Summary:**

This paper investigates prompt-based methods for continual learning. It begins by analyzing the design choices of the current two-stage inference strategy, followed by the introduction of techniques leveraging various self-supervised pretrained models for identity prediction. Specifically, a self-distillation loss is utilized to train the first-stage prediction, alongside a threshold determination strategy. Experiments are conducted on five class-incremental datasets, demonstrating improved performance in out-of-distribution (OOD) scenarios.

**Strengths:**

1. The analysis of self-supervised pretrained models for the first stage of prompt-based continual learning methods is insightful.
2. The improvement under OOD scenarios is meaningful. It is well-known that prompt-based methods performed poorly under such a setting. The solution is simple but effective.

**Weaknesses:**

1. Potential test information leakage: There is a risk of information leakage during testing when using the threshold determination. According to Section 5.2, the proposed method requires updating the parameters of the $\beta$ distribution using all test samples.
2. The empirical analysis in Section 4 does not properly support the claim made in Section 5.2. Specifically, the paper states: "Our analysis in Section 4 reveals that even if the first-stage model correctly predicts the class label, the second-stage model may still misclassify the category, despite receiving the correct task identity." However, Section 4 does not provide any results regarding the proportion of correct class label predictions by the first-stage model. It only reports task-ID prediction results, and a correct task-ID prediction does not imply a correct class label prediction.
3. Observation 2 lacks meaning: The statement "a reliable task identity benefits the second stage without requiring identical architecture for both stages" is not a significant finding. It is obvious that better task identity predictions lead to improved second-stage predictions, irrespective of architecture. A more meaningful insight might be phrased as: "Self-supervised models can provide more reliable first-stage task identity predictions than supervised models."
4. The writting of this paper is very poor. It certainly requires careful revision and polishing. Some specific issues include:
- In section 5.1.2, the parameters $\theta_{s}$ and $\theta_{t}$ are not clearly defined. It is unclear whether they represent parameters in the whole first-stage model or only the newly added parameters. If they are the added parameters, Equation 4 is problematic, as the added component does not take the input image $x$ as input. If they represent the entire first-stage model, there is a conflict with the description in line 266, which states that $\theta_{s}$ is optimized using SGD, contrary to Figure 1’s illustration.
- Symbols $K$ and $i$ are not explained in Equation 4. Does "$K$ dimensions" refer to the number of classes or tasks?
- There are inconsistencies in symbol definitions. For example, $\theta$ represents network parameters in Equation 4, but is later used to represent a distribution in line 295, Equation 5, and Equation 6. Similarly, $\tau$ is used for temperature in Equation 4 and for filtering threshold in Equation 9.
- Punctuation is missing after all equations.
- Conflict results. In table 3, the reported result for the methods with Sup-21K is 47.47, but change to 47.37 in table 5, 6, and 7.
5. Negative effect on ID datasets. Although the method shows improvements on OOD scenarios, it brings negative effect on some ID datasets. For instance, -1.06\% on CIFAR-100, -0.61\% on ImageNet-R. No ablation studies were conducted on ID scenarios, making it unclear whether both the self-distillation loss and the threshold determination strategy are harmful in these cases.

**Questions:**

See weaknesses part for questions.

---

### Official Review · Reviewer_BFhc · 2024-10-24

**Soundness:** 3
**Presentation:** 3
**Contribution:** 2
**Rating:** 5
**Confidence:** 5

**Summary:**

This paper focuses on prompt-based continual learning. An empirical analysis is firstly conducted to study the performance of supervised and self-supervised pre-trained models on ID and OOD data. The observation tells that self-supervised pre-trained models perform better on OOD data. Then a self-distillation loss is incorporated into the training process of the first-stage model. By using different models for the first stage, the accuracy can be improved further. Moreover, to avoid the misclassification raised by the second stage and reduce inference overhead, a threshold-filtering strategy is proposed to directly output the prediction made by the first-stage model. Sufficient experiments reflect that a self-supervised pre-trained first-stage model excel in OOD continual learning problem. The proposed threshold-filtering strategy can also improve the overall accuracy in OOD scenario.

**Strengths:**

1. This paper proposes a threshold-filtering strategy to reduce the unreliability and computation overhead of the second-stage model that have been ignored by previous works. It is a step towards real application.

2. The paper conducts extensive experiments to validate the presented two questions with regard to data domain and task identity inference. The observations can inspire the usage of different pre-trained backbones to handle the scenario of different domains.

3. The paper reveals that using diverse pre-trained models (especially the self-supervised models) for the first stage can benefit improving the final accuracy. As a result, the proposed method outperforms existing methods on OOD CL benchmarks.

4. The self-distillation loss shows effectiveness in improving the first-stage accuracy and thus improve the final accuracy for OOD benchmarks.

5. The paper is easy to follow.

**Weaknesses:**

1. It has been revealed by previous works [1,2] that the self-supervised technique can learn more generalizable and transferable features and thus benefit continual learning. Moreover, the self-distillation loss comes from DINO as mentioned in the Method section. Therefore, the self-supervised training for the first stage does not bring sufficiently novel insights or techniques for the community.

>[1] F. Zhu, X.-Y. Zhang, C. Wang, F. Yin, and C.-L. Liu, “Prototype Augmentation and Self-Supervision for Incremental Learning,” in IEEE/CVF Conference on Computer Vision and Pattern Recognition, Computer Vision Foundation / IEEE, 2021, pp. 5871–5880.

>[2] Z. Li et al., “Steering Prototypes with Prompt-Tuning for Rehearsal-Free Continual Learning,” in IEEE/CVF Winter Conference on Applications of Computer Vision, 2024, pp. 2523–2533.

2. The performance of the proposed method heavily depends on the pre-trained dataset and the domain/distribution of the dataset. Especially, the performance may degrade on non-OOD datasets. For example, as shown in Table 6, only the first-stage model pre-trained on DINOv2-LVD142M shows improvement on CIFAR-100, while those pre-trained on Sup-21K and DINO-1K underperform the baseline HiDe-prompt. It is essential to address the shortcomings in non-OOD scenario, since the ID data is more widely used in reality.

3. A deep analysis of the reason why the final accuracy declines (on non-OOD data) is required. In theory, the self-supervised training for the first-stage model can improve the inference accuracy of task identity. Is the threshold-filtering strategy in the second stage harmful to the overall performance?

4. In prompt-pool-based CL methods, the models of the two stages are originally decoupled and they are not enforced to be identical. Therefore, I think the first question in empirical analysis has marginal significance.

5. The threshold-filtering strategy is based on the distribution of confidence scores. The reason why the Beta distribution is adopted is not clear. Moreover, it is a straightforward and common technique, which does not provide novel ideas for a better filtering strategy.

**Questions:**

1. The accuracy of 92.35 on Split CIFAR100 in Table 4 is consistent with the accuracy in Table 6. However, why are the results for Sup-21k on the Split Aircrafts 47.47 in Table 3, but 47.37 in Tables 5 and 6?

---

### Official Review · Reviewer_T1Xf · 2024-10-27

**Soundness:** 2
**Presentation:** 2
**Contribution:** 2
**Rating:** 5
**Confidence:** 5

**Summary:**

This paper analyzes two limitations in existing methods: First, two-stage inference can still lead to errors even when reliable predictions are provided in the first stage; second, using the same architecture across both stages hinders performance improvement. To address these issues, the authors propose a threshold filtering strategy and further explore the efficiency of self-supervised pre-trained models in providing task identities.

**Strengths:**

1. The method proposed by the authors demonstrates significant performance improvement in OOD scenarios, which is consistent with their observations, proving that self-supervised pre-trained models have notable advantages in OOD settings.

2. The proposed threshold filtering strategy selectively passes data to the second stage. This approach prevents errors in the second stage when reliable predictions have already been made in the first stage, while also saving computational resources.

**Weaknesses:**

1.	The authors present results in non-OOD scenarios in the paper, where performance generally declines in most cases. However, they do not specifically analyze or explain the reasons for this performance drop. Why does performance improve significantly in OOD scenarios, while it declines in non-OOD scenarios?

2.	The experiments in the paper demonstrate that the models in the first and second stages can use different architectures and pre-trained weights, supporting the authors' findings. However, there is a lack of theoretical proof, and it is hoped that the authors will provide a detailed theoretical analysis rather than just presenting experimental results.

3.	In the paper, the authors mention, "the second-stage model may still misclassify the category, despite receiving the correct task identity. This situation occurs frequently in OOD scenarios, with an occurrence rate of around 10%." How did the authors calculate this 10% result, and is there any experimental evidence or theoretical analysis to support it?

**Questions:**

1. It is recommended that the authors provide a theoretical justification for why the models in the first and second stages can use different architectures and pre-trained weights.

2. It is suggested that the authors conduct a deeper analysis of why performance improvements are not significant in non-OOD scenarios, rather than simply listing experimental results.

---

### Official Review · Reviewer_w7DN · 2024-11-09

**Soundness:** 2
**Presentation:** 3
**Contribution:** 2
**Rating:** 3
**Confidence:** 4

**Summary:**

The topic of this paper is prompt-based Class-Incremental Learning (CIL) and the authors give a summary of key issues in two-stage prompt-based CIL methods. To alleviate these issues, they introduces self-supervised learning, a threshold filtering strategy and self-supervised pre-trained models at the first-stage and second stage. The experiments are conducted on both the In-Distribution(ID) and Out-Of-Distribution(OOD) CIL benchmarks.

**Strengths:**

+ The paper is well-written and easy to follow.
+ The proposed method is simple and achieves a performance gain on several OOD CIL benchmarks.
+ Exploring the prompt-based CIL at OOD CIL scenarios is interesting

**Weaknesses:**

- Some of prompt-based CIL methods dynamically generate prompts, and these important works are not discussed and compared in this paper. As described in this paper, the two-stage “select+prediction” prompt-based CIL methods usually meet the issues at the first stage. However, some “prompt-generation+prediction” prompt-based CIL methods (e.g. [a]-[b]) directly generate instance-wise prompts and do not need to select prompts at the first stage. It is necessary to give a detailed analysis about these methods. For instance, compare the performance and memory cost of these methods on both the ID and OOD CIL benchmarks.

[a] Generating Instance-level Prompts for Rehearsal-free Continual Learning, ICCV2023

[b] Open-World Dynamic Prompt and Continual Visual Representation Learning, ECCV2024

- The uniqueness of the summary on prompt-based methods is unclear. Previous works (e.g. [c]-[d]) have made a summary about prompt-based CIL methods and it makes the contribution of this paper limited. To better show the unique contribution of this paper, the key difference among these works should be highlighted. The authors can provide performance comparison experiments on ID and OOD scenarios to show the necessity of combining self-supervised learning to prompt-based CIL methods.

[c] Continual Learning with Pre-Trained Models: A Survey, IJCAI2024

[d] Revisiting Class-Incremental Learning with Pre-Trained Models: Generalizability and Adaptivity are All You Need, arxiv2023

- Missing the experimental analysis on some complex CIL scenarios (e.g. on some challenging CIL benchmarks). For example, the work[d] has proposed four new challenging ID CIL benchmarks without data overlapping. To better show the effectiveness of the proposed method, the experimental analysis on some complex CIL benchmarks are needed (e.g., evaluate these methods on the cross-domain CIL benchmark VTAB [d]).

- To my knowledge, the effectiveness of utilizing a self-supervised pre-trained model to provide task-ids depends on the gap between the pre-trained data and incrementally learned tasks. If the gap is large, a self-supervised pre-trained model can not provide reliable task identities. Can the authors make a discussion about this point? Compare the performance of the proposed method across CIL datasets with varying degrees of similarity to pre-trained data.

**Questions:**

My major concerns come from the experimental analysis, comparison to related works, the unique contribution and the claim of the task-id prediction module. The details are included in the above weaknesses.

---

### Note · Authors · 2024-11-14

I have read and agree with the venue's withdrawal policy on behalf of myself and my co-authors.